# OpenReview forum: "Multilingual Jailbreak Challenges in Large Language Models"
_ICLR.cc/2024/Conference — ICLR 2024 poster_

### Official Review · Reviewer_WaDP · 2023-10-31

**Soundness:** 3 good
**Presentation:** 3 good
**Contribution:** 2 fair
**Rating:** 6
**Confidence:** 3

**Summary:**

In this paper, the authors identify some of the challenges regarding the presence of multilingual jailbreak in two different scenarios: intentional and unintentional. Moreover, they introduce a multilingual jailbreak dataset, analyze the effects of language as a jailbreak method, and show that medium and low-resource languages are more likely to generate unsafe content compared to high-resource ones. Finally, they propose a framework called SELF-DEFENCE, which starts from a set of English seed input-output pairs, including both unsafe and general examples, and augments the dataset using the LLM and these seed examples. Then, it uses the LLM to translate the instruction pairs into multiple target languages and merges these language-specific corpora to form the final training dataset for fine-tuning. Finally, the authors use the fine-tuning access for ChatGPT to evaluate the effectiveness of their framework. Their results show improvements in the multilingual safety capabilities of LLMs.

**Strengths:**

- The authors introduce a multilingual jailbreak dataset called MultiJail, which they claim is the first multilingual jailbreak dataset. The process involves incorporating native speakers for human translations in order to prevent noisy translation.
- Section 3 provides a detailed evaluation of the multilingual safety challenges of ChatGPT and GPT-4 and provides insight into the effects of language in two cases, intentional and unintentional.

**Weaknesses:**

- The data generated for finetuning is mostly generated by the LLM itself (except for the small number of seed examples). While this makes the data generation easier and cheaper, I assume the translations by the LLM for the low-resource languages are still very noisy and might be the reason why usefulness reduces significantly as the safeness metric increases. The authors mention that a potential reason for this decrease in general capability can be attributed to the fact that LLM rejects to answer unsafe questions. In that case, it would be useful to have a numerical comparison between the cases that get low usefulness scores because of this issue and the cases that fail due to noisy translations. As the authors mention, it would be useful to include a brief explanation of why the question is unsafe to ensure the problem is not, in fact, due to noisy translations of the low-resource language by the LLM.
- In my opinion, the SELF-DEFENCE framework lacks novelty. It simply implies that providing more training samples for lower resource languages can improve LLM's understanding of them, and as we include samples translated to these languages during the fine-tuning stage, the model becomes safer with respect to them.
- There is no information on the effectiveness of the SELF-DEFENCE framework compared to any other safety improvement techniques mentioned in the related work section.

**Questions:**

- How does the SELF-DEFENCE framework perform compared to other safety improvement methods, such as RLHF?
- For low-resource languages, where the LLM's understanding of the language is limited, how reliable are their LLM-generated translations?

---

> ### Author Response · Authors · 2023-11-20
>
> Thank you so much for taking the time to share your valuable feedback with us. We truly appreciate your insights, and please find below our responses to your concerns:
>
> >**W1. Noisy translations by LLM may reduce usefulness**
>
> Thank you for pointing this out. Yes, we agree with you that noisy translation may be another cause for usefulness reduction. To verify this assumption, we replaced the translation component in SELF-DEFENCE to Goolge Translate, which is believed to perform better than ChatGPT on machine translation [1].  The results are presented as below:
>
> |            | Safeness | Usefulness |
> |------------|----------|------------|
> | SELF-DEFENCE | 63.50    | 41.39      |
> | Google-Translate | 64.50    | 44.44      |
>
> With better translation quality, both safeness and usefulness have gained improvement, especially for usefulness. We have modified our paper accordingly to point this factor out.
>
> However, we believe a less comprehensive safety Q&A  would also be an important factor, as the trade-off is also observed even in English-only evaluation [2].
>
> In conclusion,  the quality of generated content plays a crucial role, both in terms of language-wise and the content itself. In our future research, we aim to delve into the enhancement of the quality of data generated by Self-Defence framework. Again, thank you very much for your valuable feedback.
>
> >**W2. SELF-DEFENCE framework lacks novelty**
>
> We think providing more training examples for low-resource languages can improve LLM's multilingual safety capabilities. However, our proposed method can be helpful in the following scenarios:
>
> 1. Data augmentation: Our method provides a way to quickly augment a large multilingual safety dataset to address the challenge effectively and efficiently.
> 2. Fine-grained generation: If a new safety topic is introduced, or the safety guideline is changed, or a new language is added, we can simply edit the seed example and instruction in the "Self-defense" framework to adapt accordingly.
> 3. Data imbalance: Safety datasets may have uneven topic distribution. The "Self-Defense" framework can address this by generating data for underrepresented topics using specific seed examples and instructions.
> 4. Avoiding intellectual property conflict: As data becomes more and more important, the safety dataset may have conflicts of interest or be regulated by licenses. However, utilizing "Self-defense" can easily avoid them and generate data by the LLM itself.
>
> In conclusion, while directly augmenting existing datasets with non-English data is a viable approach, our method introduces flexibility, adaptability, and a way to circumvent potential legal issues. This makes it a valuable tool in the evolving landscape of AI safety.
>
> >**W3. No comparison of SELF-DEFENCE with other safety improvement techniques**
>
> Thank you for your feedback. It's important to note that our proposed SELF-DEFENCE framework should be viewed as complementary, rather than directly comparable, to existing safety improvement techniques such as SFT and RLHF. Unlike these methods which focus on safety improvement through training algorithm design, the SELF-DEFENCE approach primarily serves to augment multilingual safety datasets. Our approach can be combined with other safety improvement techniques for more comprehensive results. In fact, during our experimentation, we employed the SFT method to exploit the augmented dataset provided by our SELF-DEFENCE framework for safety training.
>
> For comparison, we conducted another experiment using English-only data, which is the most common language in safety tuning, as a baseline. The following table presents the experimental results:
>
> |              | Safeness | Usefulness |
> |--------------|----------|------------|
> | SELF-DEFENCE | 63.50    | 41.39      |
> | English-only | 60.67    | 37.78      |
>
> As the results show, SELF-DEFENSE can not only improve multilingual safety capability, but the multilingual usefulness is also maintained. The results demonstrated the effectiveness of our proposed framework.
>
> >**Q1. Performance of SELF-DEFENCE vs other safety improvement methods**
>
> We have addressed this concern in W3.
>
> >**Q2. Reliability of LLM-generated translations for low-resource languages**
>
> We have addressed this concern in W1.
>
> [1] Jiao et al, Is ChatGPT A Good Translator? Yes With GPT-4 As The Engine, https://arxiv.org/abs/2301.08745
>
> [2] Bai et al, Training a helpful and harmless assistant with reinforcement learning from human feedback, https://arxiv.org/abs/2204.05862

---

> ### Author Response · Authors · 2023-11-22
> **More discussion with Reviewer WaDP**
>
> Dear Reviewer WaDP,
>
> We thank you for the time spent reviewing our work and for your constructive comments. We sincerely hope to have further discussions with you to see if our responses address your concerns.
>
> Specifically, in our response, we have:
>
> 1. Clarified how noisy translation affects our work;
> 2. Presented the novelty of our Self-Defence method;
> 3. Compared our work to other techniques.
>
> We genuinely hope you will review our responses. Thank you!
>
> Best Regards,
>
> Authors

---

> > ### Comment · Reviewer_WaDP · 2023-11-23
> >
> > Thank you for providing such detailed explanations and additional experiments. I have changed my overall score to 6.

---

### Official Review · Reviewer_imbS · 2023-10-31

**Soundness:** 3 good
**Presentation:** 3 good
**Contribution:** 3 good
**Rating:** 6
**Confidence:** 3

**Summary:**

The paper shows that the safeguard ChatGPT and GPT-4 are not robust for non-English languages, especially low resource languages. By translating curated harmful prompts in English and translating them into 9 other languages, the paper shows the increase of unsafe rates in the responses of LLMs. To this end, the paper proposes a self-defence method that generates finetuning data for chatGPT and a tuned version of chatGPT on this data exhibits a lower unsafe rate.

**Strengths:**

- The methodology of collecting harmful prompts, running human translation, and using human evaluation for the 15 harmful prompts in the preliminary results. I also like the approach of using jailbreakchat.com for collecting intentional jailbreak prompts.


- Overall, the paper is well motivated and covers interesting analysis such as the translation quality, jailbreak languages, the trade off between safeness and usefulness.

**Weaknesses:**

- Lack of details of the self-defence algorithm. What are the seed samples and what is the percentage of unsafe samples generated by chatGPT that are actually unsafe if it is prompted to chatGPT to get the answer?

- How wide-coverage in terms of topics, scenarios,...of the data that the self-defence algorithm can generate? While it’s good to show some improvement when evaluated on 	MultiJail testset, I think self-defence just scratches the surface of the problem.

- I generally think it’s important to ensure safety for all the languages but it irks me about the argument about Bengali at the end of page 5th. Even though there are 285 million native speakers, how many of those are using chatGPT and could be affected by unsafe outputs?

**Questions:**

See the question in the above section.

---

> ### Author Response · Authors · 2023-11-20
>
> We greatly appreciate the time you dedicated to providing us with your valuable feedback. Your insights are highly valued, and please find our responses to the concerns you raised below:
>
> >**W1. Lack of details on self-defence algorithm**
>
> The seed examples utilized within our experiment are crafted by humans to simulate both unintentional and intentional scenarios. We supply no more than six seed examples in the generation process.
>
> When it comes to the generation of unsafe examples, we instruct the model with generating these in English. The model's strong proficiency in English and its robust safety capabilities within this language allow it to produce content that is genuinely unsafe. To ensure the validity of this process, we have each generated example manually reviewed and verified to confirm its unsafe content.
>
> >**W2. Unclear coverage of self-defence algorithm's data generation**
>
> In our current work, we have provided a set of general instructions for generating safety-related Q&A pairs without delving into specific topics or scenarios. However, our "Self-Defense" method is flexible and can be easily expanded to more advanced approaches.
>
> As an example, we might supply the LLMs with several pre-defined safety topics, or even enable the model to generate these topics by themself. For each distinct topic, we could gather or generate relevant seed examples, and adjust the instructions to match the topic accordingly.
>
> While the current focus is on this immediate concern, our future work aims to refine and enhance the "Self-Defense" methodology, thereby improving its effectiveness and efficiency in addressing multilingual jailbreak challenges.
>
> >**W3. Questioning relevance of safety measures for low-use languages like Bengali**
>
> Thank you for your valuable input. We totally agree that the number of native speakers doesn't necessarily imply a significant impact on the usage of ChatGPT. However, our mention of Bengali aimed to highlight safety issues in low-resource languages, which refers to the scarcity of available digital training data, not the speaker count.
>
> Despite being a low-resource language in AI training, Bengali is the 6th most spoken language worldwide with 285 million native speakers [1]. This significant speaker base often gets overlooked in AI safety considerations, which typically focus on languages with abundant digital resources.
>
> While Bengali-speaking ChatGPT users may be a small fraction of the total user base, it's crucial to note that Bengali is one among many low-resource languages. Cumulatively, these languages represent a substantial user base, emphasizing the need for dedicated safety measures.
>
> Thus, our discussion extends beyond Bengali to all low-resource languages, many with large speaker populations. We'll clarify this in future discussions and appreciate your insights in improving our communication.
>
> [1] List of languages by number of native speakers, https://en.wikipedia.org/wiki/List_of_languages_by_number_of_native_speakers

---

> ### Author Response · Authors · 2023-11-22
> **More discussion with Reviewer imbS**
>
> Deaer Reviewer imbS:
>
> We appreciate you taking the time to review our work and your insightful feedback. We would like to discuss this further with you to see if there are any unresolved questions. Specifically, we have addressed your concerns on the following points.
>
> 1. More details on Self-Defence;
> 2. Discussion on coverage of Self-Defence;
> 3. Relevance for low-resource languages.
>
> If you have any further concerns, please let us know.
>
> Best Regards,
>
> Authors

---

### Official Review · Reviewer_QBvZ · 2023-11-03

**Soundness:** 3 good
**Presentation:** 4 excellent
**Contribution:** 3 good
**Rating:** 6
**Confidence:** 4

**Summary:**

In this paper the authors explore how LLM safety generalizes across languages.  They translate a set of unsafe prompts across multiple languages, finding that ChatGPT and GPT-4 safety metrics degrade by language prevalence and that the models are also more vulnerable to a jailbreak when combining languages.  They propose additional fine-tuning data to improve safety across these languages, finding that this fine-tuning improves safety but comes with a cost to usefulness.

**Strengths:**

S1. The question of safety across many languages and generalization of LLMs across languages is of significant importance to this technology.

S2. There has been some but not a lot of work so far doing a thorough evaluation of LLM safety across languages, making this evaluation valuable for the community.

S3. The paper studies this question from numerous angles - overall safety performance, jailbreaks, amount of data in CommonCrawl, mitigation through fine-tuning, trade-off with usefulness.

**Weaknesses:**

W1. I think for the study of safety in other languages, it should not be framed as a jailbreak.  Simply ensuring the model performs safely for people who speak those languages is important on its own.  In some places this acknowledged but I'd make it more clear early on and consistent.

W2. Small sample size and error bars - samples are generally quite small (15-30 samples per language in some cases; only testing on a single jailbreak, AIM).  It'd be good to see this work done at a larger scale (especially given much of it is automated) and to include error bars given the small sample sizes.

W3. Also given the small sample size, it is hard to gauge what diversity of safety issues are studied.  More details on the types of safety issues and how those vary across language would help contextualize the results (although I believe this is less critical).

W4. Experiments are only run on ChatGPT and GPT-4.  It'd be useful to see similar experiments on other models, e.g. through HELM or just a few other APIs (again not critical but valuable).

W5. The method seems somewhat over-complicated, when the data eg from Anthropic could be translated and used directly.  How come this more complex algorithm is necessary/valuable?

W6. Overall I'd consider the mitigation section the weakest section - it'd be nice to see the authors find a way to improve safety without hurting usefulness, and to understand the benefits of additional data with as much nuance as the earlier sections (eg results don't seem to be broken down here by level of language resources anymore).

**Questions:**

While W1 is mostly a writing suggestion, further experimental details and answers for W2-W6 above would be appreciated.

---

> ### Author Response · Authors · 2023-11-20
>
> Thank you for providing us with your valuable feedback and suggestions. We appreciate your input and have carefully considered your questions. Below, we provide detailed responses to each of them:
>
> >**W1: Frame safety study as important, not as a 'jailbreak'**
>
> Thank you for your valuable feedback. While we agree that ensuring safety for different languages is important in itself, we wanted to provide a novel perspective by highlighting how language can be used as a method for jailbreaking. This framing allows us to explore the vulnerabilities of LLMs across languages. We will revise the paper to ensure that the importance of language-specific safety is emphasized alongside the concept of jailbreaking.
>
> >**W2: Small sample size and lack of error bars**
>
> **Regarding the small sample size**, we included 15 examples in our preliminary experiment as a starting point. We do recognize the need for a larger dataset to obtain more reliable results. Therefore, in the detailed experiment, we expanded our dataset to include 315 examples for each language, resulting in a total of 3150 examples.
>
> **Regarding the malicious instructions**, our intention was to replicate the actions of a real-life malicious user who searches for and selects the most powerful malicious instructions available online. That't why the scenario is named as "intentional". While we acknowledge that including a greater variety of malicious instructions would provide a more comprehensive evaluation, we plan to incorporate additional instructions in our future research.
>
> **Regarding the error bars**, given that we have set the temperature to 0 and are evaluating a fixed dataset, the output remains constant. To respond to your concerns, we have also conducted an additional evaluation using another decoding method. Specifically, we used nucleus sampling with a top_p value of 0.8. To ensure reliable results, we ran the experiment three times with different seeds. The following data represents the unsafe rate of this experiment, with the standard deviation values indicated within parentheses:
>
> |   | en | zh | it | vi | HRL | ar | ko | th | MRL | bn | sw | jv | LRL | Avg. |
> |---|----|----|----|----|-----|----|----|----|-----|----|----|----|-----|------|
> | ChatGPT (sample=False) | 0.63 | 2.22 | 2.86 | 7.94 | 4.34 | 6.03 | 9.84 | 18.10 | 11.32 | 28.25 | 7.94 | 8.57 | 14.92 | 10.19 |
> | ChatGPT (nucleus sampling with top_p = 0.8) | 0.42 (0.18) | 4.02 (0.48) | 2.75 (0.37) | 9.10 (0.48) | 5.29 (0.21) | 6.88 (0.48) | 9.84 (0.84) | 20.95 (1.45) | 12.56 (0.34) | 31.85 (1.28) | 8.15 (1.20) | 9.42 (1.43) | 16.47 (0.60) | 11.44 (0.31) |
>
> It is evident that the average unsafe rate, although slightly higher (1.25%), the trend is still clearly observable, as the rate of unsafe content increases with decreasing language availability, resulting in a consistent ranking order. We have show details in Appendix A.6.
>
> >**W3: Need for diversity in safety issues studied**
>
> Yes, we have incorporated tag features to indicate the safety issue studies. These tags are derived from the Anthropic dataset, which assigns one or more tags to each example. In order to maintain consistency, we have applied the same tagging schema to label OpenAI's 15 prompts. Our dataset includes a wide range of 18 safety issues, ensuring its diversity. The top-3 safety issues and their respective percentages in the dataset are "Violence & incitement" (18.70%), "Discrimination & injustice" (11.28%), and "Hate speech & offensive language" (8.41%). For a comprehensive list of tags and their details, please refer to the Appendix A.2.

---

> ### Author Response · Authors · 2023-11-20
>
> >**W4: Tests limited to ChatGPT and GPT-4**
>
> We expanded our experiment to include two widely used open-source LLMs: Llam2-Chat and Vicuna. The results of this extension on unintentional scenario are presented below:
>
> | | ChatGPT | GPT-4 | Llama2-7b-Chat | Vicuna-7b-v1.5 |
> |---|---|---|---|---|
> | en - unsafe | 0.63 | 0.95 | 0.63 | 57.14 |
> | en - safe | 99.37 | 99.05 | 99.37 | 37.78 |
> | en - invalid | 0.00 | 0.00 | 0.00 | 5.08 |
> | HRL - unsafe | 4.34 | 3.60 | 2.22 | 40.32 |
> | HRL - safe | 95.13 | 95.98 | 92.06 | 51.32 |
> | HRL - invalid | 0.53 | 0.42 | 5.71 | 8.36 |
> | MRL - unsafe | 11.32 | 4.13 | 4.55 | 42.75 |
> | MRL - safe | 87.20 | 94.94 | 66.88 | 32.38 |
> | MRL - invalid | 1.48 | 0.95 | 28.57 | 24.87 |
> | LRL - unsafe | 14.92 | 10.16 | 1.69 | 28.78 |
> | LRL - safe | 78.41 | 83.49 | 65.19 | 9.42 |
> | LRL - invalid | 6.67 | 6.35 | 33.12 | 61.80 |
> | AVG - unsafe | 14.92 | 5.96 | 2.82 | 37.28 |
> | AVG - safe | 78.41 | 91.46 | 74.71 | 31.04 |
> | AVG - invalid | 6.67 | 2.57 | 22.47 | 31.68 |
>
> However, we found that these two models contain the following problems:
>
> 1. Although Llama2-Chat has the capacity to comprehend non-English languages, it mostly responds in English. This may limit its applicability in real-world scenarios, particularly for non-English speaking users, making the evaluation less realistic.
> 2. Both models often generate invalid outputs due to their limited multilingual capabilities. While Llama2-Chat owns the lowest average rate of unsafe content, it is challenging to determine whether this lower rate is a consequence of producing genuinely safe content or simply generating invalid content.
> 3. Vicuna has been trained on user-shared conversations sourced from ChatGPT and GPT-4. The language distribution within this training data is somewhat disordered, leading to unpredictable results. Furthermore, Vicuna has not undergone specific safety tuning, resulting in a high unsafe content rate, particularly in English, with a staggering 57.17% unsafe rate.
>
> Thus, we have decided to concentrate our evaluation primarily on ChatGPT and GPT-4 to ensure a realistic and dependable assessment. However, we will supply the supplementary results in the Appendix A.7 for further reference.
>
> >**W5: Over-complicated methodology**
>
> We acknowledge that directly translating Anthropic's dataset or other safety-related datasets is a straightforward way to address the multilingual jailbreak challenge. However, our proposed method can be helpful in the following scenarios:
>
> 1. Data augmentation: Our method provides a way to quickly augment a large multilingual safety dataset to address the challenge effectively and efficiently.
> 2. Fine-grained generation: If a new safety topic is introduced, or the safety guideline is changed, or a new language is added, we can simply edit the seed example and instruction in the "Self-defence" framework to adapt accordingly.
> 3. Data imbalance: Safety datasets may have uneven topic distribution. The "Self-Defence" framework can address this by generating data for underrepresented topics using specific seed examples and instructions.
> 4. Avoiding Intellectual property conflict: As data becomes more and more important, the safety dataset may have conflicts of interests or be regulated by licenses. However, utilizing "Self-defence" can easily avoid them and generate data by the LLM itself.
>
> In conclusion, while the direct translation of existing datasets is a viable approach, our method introduces flexibility, adaptability, and a way to circumvent potential legal issues. This makes it a valuable tool in the evolving landscape of AI safety.
>
> >**W6: Weak mitigation section, needs improved safety-usefulness balance**
>
> Thank you for your kind advice. We believe that more fine-grained instructions for data generation could lead to more detailed and comprehensive responses. This approach could prevent the model from merely learning to reject from the safety training data, thereby enhancing the model's ability to maintain high usefulness while ensuring safety. We are eager to explore this aspect further in our future research.
>
> Regarding a more nuanced understanding of the benefits of additional data, we present the unsafe decrease rate by language category as below:
>
> | | HRL | MRL | LRL |
> |---|---|---|---|
> | unintentional | 3.60 | 6.56 | 8.57 |
> | intentional | 13.86 | 24.23 | 24.66 |
>
> In both scenarios under consideration, it's evident that LRL achieves the most significant improvement. This could potentially be attributed to its initially high rate of unsafe content, implying that even a small amount of training data can yield substantial gains.
>
> Furthermore, we observe that the intentional scenario demonstrates a more pronounced improvement compared to the unintentional one. This suggests a greater potential for enhancement to mitigate intentional jailbreak attack.

---

> ### Author Response · Authors · 2023-11-22
> **More discussion with Reviewer QBvZ**
>
> Dear Reviewer QBvZ:
>
> Thanks so much for your valuable suggestion on:
>
> 1. Manuscript frame;
> 2. Error bar;
> 3. Diversity;
> 4. More model evaluation;
> 5. Methodology motivation;
> 6. Trade-off analysis.
>
> We have carefully responded to your feedback and conducted supplementary experiments to address your concerns. We are eagerly anticipating your response and thoughts on our new experiments as the deadline for discussion draws near.
>
> Best Regards,
>
> Authors

---

> > ### Comment · Reviewer_QBvZ · 2023-11-22
> > **Thanks!**
> >
> > Thank you for your detailed responses, and I especially appreciate the additional experimental results in the short time frame.    A few small clarifications:
> >
> > With respect to W2:
> > - To clarify, my question on diversity of attacks - it is not clear to me that an adversary would know a-priori which attacks are best, so understanding the breadth of attacks is valuable.
> >
> > - For error bars, changing the temperature does add uncertainty but my concern was with the error bars from a small sample (eg could be measured with bootstrap sampling)
> >
> > For W3: Thanks for the breakdown.  While it would require a lot more work, it'd be nice to see results broken down by these tags (which of course requires more data per tag)
> >
> >
> >
> > Overall, I still believe the paper would be valuable to the community but also has room for improvement. As a result, I will keep my rating as is. Thank you.

---

> > > ### Author Response · Authors · 2023-11-23
> > > **Reply to Reviewer QBvZ**
> > >
> > > Dear Reviewer QBvZ,
> > >
> > > Thank you for your response! We have addressed your concerns as follows:
> > >
> > > >**W2. Additional Attack**
> > >
> > > 1. Regarding more attacks:
> > >
> > > As mentioned in footnote 4 on page 5, a "Votes" score exists that could potentially offer adversaries a basic understanding of which attacks are most effective.
> > >
> > > We acknowledge the importance of incorporating a wider range of attacks into our analysis. Although our initial focus was primarily on the language level, we are keen to broaden our investigation scope in future research. Furthermore, there have been several studies evaluating general jailbreak methods, which may serve as useful references [1,2].
> > >
> > > 2. Regarding error bars:
> > >
> > > Thanks for your clarification! We will improve our manuscript to make our preliminary experiment more reliable.
> > >
> > > >**W3. Break down by tags**
> > >
> > > We have revised our manuscript and present the outcomes in Appendix Figure 8. It's evident that different languages exhibit varying levels of safety performance depending on the specific tag. For instance, posing queries about weapons in Bengali to ChatGPT demonstrates a notably higher unsafe rate compared to other languages. Similarly, when interacting with ChatGPT in Thai about substance abuse, the unsafe rate is significantly higher in comparison to other languages. These observations underline the potential vulnerabilities and biases intrinsic to each language within the model. Such findings emphasize the need for continuous improvements and targeted refinement in the model's safety capabilities across different languages.
> > >
> > > Overall, it was a nice discussion, and your valuable suggestions have helped us a lot to improve our manuscript. We deeply appreciate your time and effort. Thank you!
> > >
> > > Best Regards,
> > >
> > > Authors
> > >
> > >
> > > [1] Deng et al, Jailbreaking ChatGPT via Prompt Engineering: An Empirical Study, https://arxiv.org/abs/2305.13860
> > >
> > > [2] Wei et al, Jailbroken: How Does LLM Safety Training Fail?, https://arxiv.org/abs/2307.02483

---

### Official Review · Reviewer_Pbxw · 2023-11-04

**Soundness:** 3 good
**Presentation:** 2 fair
**Contribution:** 3 good
**Rating:** 6
**Confidence:** 4

**Summary:**

This paper proposes multilingual jailbreaks in two scenarios, 1) the unintentional scenario involves users querying LLMs using non-English prompts and inadvertently bypassing the safety mechanisms, 2) the intentional scenario entails malicious users combining jailbreak instructions with multilingual prompts to attack LLMs deliberately. Empirical experiments show that proprietary models are vulnerable in unintentional scenarios. Besides, a self-defense strategy is proposed where both harmful and harmless prompts and desired responses in different languages are utilized to fine-tune chatgpt.

**Strengths:**

1. This paper proposes an effective jailbreak strategy, with detailed analysis on a defense attempt as well.
2. Two jailbreak scenarios are investigated, one is intentional and the other is unintentional.

**Weaknesses:**

1. Using output from GPT4 as evaluation measure is not a good idea although its output looks consistent as human evaluation demonstrated in Fig.2

As shown in Table 1, GPT-4 still suffers from jailbreak when the user prompt is in English: as high as 28.25% for unintentional case.

2. It's hard to distinguish the high success attack rate is attributed to rarely unseen language or the AIM harmful instruction, since the harmful instruction itself alone can achieve very high success attack rate in English, as shown in Table 1.

**Questions:**

In section 3.1. Setup, Since Anthropic's red teaming dataset contains multi-turn dialogs, it's likely that the first user prompt is harmless. Perhaps you should consider both task_descripton_harmlessness_score and the number of turns (turn=1 to ensure that the 1st user prompt is harmful if the score satisfies some pre-defined criterion). Moreover, there are 38961 instances covered by the red teaming dataset, the details for how to select 300 from 38k instances should be elaborated.

**Suggestion**

In tables, all the unsafe rate values are presented in percentile. But in figures, values are shown in decimal. A consistent presentation throughout the paper is preferred for easier understanding.

---

> ### Author Response · Authors · 2023-11-20
>
> Thank you for providing us with your valuable feedback. We have carefully considered your questions and would like to address them as below:
>
> >**W1. Concern on GPT-4 evaluation**
>
> Regarding safety evaluation, it's worth highlighting that it's generally an easier task compared to generating safe content. The former is a classification task, while the latter is a more complex natural language generation task. A recent study provides further evidence of this, showing that LLMs can effectively flag a majority of harmful text they generate themselves [1].
>
> In addition, the table below summarizes common methods for evaluating jailbreak results, their limitations, and how GPT-4 compares to them:
> | Evaluation Method | Description | Limitations | Comparison with GPT-4 |
> |-------------------|-------------|-------------|-----------------------|
> | Rule Match | Rule-based method checking if a response contains rejection words (e.g., "I'm sorry", "I cannot") | Poor performance and cannot handle "invalid" class. | GPT-4 is stronger and can handle "invalid" cases better |
> | Moderation API | Utilizes OpenAI/Google's moderation APIs to get scores for pre-defined unsafe classes | Poor performance, only contains  pre-defined unsafe classes, cannot handle "invalid" class | GPT-4 performs better and can handle more classes, including the "invalid" class |
> | Fine-tuned model | Fine-tune a small language model for evaluation | Lack of reliable training datasdet. | GPT-4 can execute evaluations in a zero-shot manner, delivering impressive results. |
> | Human Evaluation | Uses humans as evaluators | Costly, inefficient, and can be subjective, leading to inconsistent scores | GPT-4 is more cost-effective, efficient, and provides consistent scoring |
>
> Taking into account the aforementioned factors, GPT-4 has emerged as a reliable, efficient and widely-accepted evaluation method [2,3,4,5].
>
> Moreover, as depicted in Fig. 2, there is a significant level of agreement between human evaluation and the evaluation carried out by GPT-4. To quantify this agreement, we computed the Cohen's kappa score, which stands at a robust 0.86. This high score underscores the considerable alignment between GPT-4's evaluations and human judgments.
>
> Based on these reasons, we have made the decision to employ GPT-4 as our primary evaluation measure.
>
> >**W2. Attribution of high attack success rate**
>
> Yes, we intentionally chose the highest malicious instruction from https://www.jailbreakchat.com/, but it is crucial to clarify that our primary objective wasn't to obtain a high unsafe rate. Instead, we attempted to **mimic a malicious user's behavior** who, in a real-life scenario, would likely search the internet to find the most effective harmful instruction for intentional purposes.
>
> We wanted to demonstrate that even when a malicious user employs the most advanced malicious instruction, the integration of multilingualism can substantially enhance the effectiveness of their jailbreaking techniques.
>
> Therefore, the high unsafe rate should not be solely attributed to the malicious instruction power. Instead, it proves how multilingualism can amplify the potential of already powerful tools, thereby posing a significant threat to AI systems.
>
> By incorporating multilingualism, we are exploring an orthogonal approach to boosting jailbreaking techniques. It is a unique dimension that can amplify the threat level, thereby highlighting the necessity for robust multilingual safety measures.
>
> >**Q1. Clarification on setup from red teaming dataset**
>
> Below are the details of our data selection process:
> 1. We selectively choose adversarial examples that include tags. Out of a total of 38,961 examples, only 742 of them contain harmful tags.
> 2. From this set, we randomly sample 300 examples from top-500 lowest task_description_harmlessness_score. All of these examples have both harmful tags and a low task_description_harmlessness_score. We then extract the first user prompt from each of them.
> 3. Following a human evaluation, it was determined that 99% (297/300) of the extracted examples were harmful.
>
> We have included the detailed tag statistics in Appendix A.2.
>
> >**Q2. Consistency in value presentation between tables and figures**
>
> Thank you for your kind suggestion! We will refine accordingly to maintain consistent presentation.
>
> [1] Phute et al., LLM Self Defense: By Self Examination, LLMs Know They Are Being Tricked, https://arxiv.org/abs/2308.07308
>
> [2] Yuan et al, GPT-4 Is Too Smart To Be Safe: Stealthy Chat with LLMs via Cipher, https://arxiv.org/abs/2308.06463
>
> [3] Bhardwaj et al, Red-Teaming Large Language Models using Chain of Utterances for Safety-Alignment, https://arxiv.org/abs/2308.09662
>
> [4] Qi et al, Fine-tuning Aligned Language Models Compromises Safety, Even When Users Do Not Intend To!, https://arxiv.org/abs/2310.03693
>
> [5] Ji et al, BeaverTails: Towards Improved Safety Alignment of LLM via a Human-Preference Dataset, https://arxiv.org/abs/2307.04657

---

> ### Author Response · Authors · 2023-11-22
> **More discussion with Reviewer Pbxw**
>
> Dear Reviewer Pbxw,
>
> We greatly appreciate your valuable comments on our work. We have carefully addressed your concerns regarding:
>
> 1. Reason for GPT-4 evaluation;
> 2. Attribution of high attack success rate;
> 3. Clarification on dataset setup.
>
> As the deadline for discussion is approaching, we eagerly await your feedback on our response and new experiments.
>
> Thank you for your time and consideration.
>
> Best Regards,
>
> Authors

---

### Official Review · Reviewer_emGp · 2023-11-06

**Soundness:** 2 fair
**Presentation:** 2 fair
**Contribution:** 3 good
**Rating:** 8
**Confidence:** 4

**Summary:**

This paper reports the vulnerabilities of large language models from the perspective of two scenarios: unintentional and intentional situations. In the unintentional scenario, queries that are simply translated into non-English languages unexpectedly make users face unsafe content. On the other hand, the intentional scenario considers translated multi-lingual “jailbreak” prompts. The paper evaluates ChatGPT and GPT-4, and reveals that the models are more risky as the target language is lower-resource. Lastly, they propose “Self-Defense” method that fine-tunes ChatGPT with augmented and translated unsafe data.

**Strengths:**

This paper addresses timely and crucial issue of LLMs, as those models are usually evaluated in English, despite they have been widespread around the world. I think we should pay attention to strengthening the safety level of LLMs for the various languages including low-resource languages. In that sense, the multilingual jailbreak and the evaluation results are valuable.

**Weaknesses:**

Although the significance of the theme of this paper, in overall, the soundness and presentation are quite needed to be improved to strengthen this paper. Here are my comments and feedback. Some might be nice-to-haves but not strictly necessary, so feel free to argue why it is a good idea not to incorporate them.

1. Overall, when you mention differences or improvements in performance, please mention the significance levels (ie., p-values) together. Also, please denote standard deviations (inside tables or figures), because the number of test samples is limited (15 prompts for the preliminary experiment).
2. Lack of human evaluation details. Moreover, decisions on whether generated content is safe or unsafe are subjective and depend on annotators. The paper should include the human evaluation process, instructions, and annotator information.
3. Recently, there has been an argument that the evaluation of model-generated output with LLM-based evaluators has a potential bias — that is, GPT-4 might prefer and give higher scores on GPT-4’s outputs than ChatGPT [1, 2]. Also, (in Table 1), GPT-4 might have been aligned more than ChatGPT concerning GPT-4’s safety evaluation standards. Moreover, GPT-4 as an evaluator might have led to length or positional bias [2]. The paper should mention this if you have already considered this, or reinforce the results with repetitive experiments.
4. Moreover, in Section 2, the authors mentioned that “Furthermore, Figure 2 illustrates a notable level of agreement between human annotators and the GPT-4 evaluator.”, I think you can provide the correlation or agreement values (numbers) rather than just pointing out similar trends of the two lines.
5. Release of MultiJail dataset and evaluation results. — I believe the authors would be able to publish their data.
6. This paper evaluated only two black-box models, whose training corpus and portion of languages are unveiled. What about evaluating other open-sourced models such as LLaMA, Vicuna, and others, to see the correlation between the portions of language resources in training data and the attack success rates?
7. The proposed “Self-Defense” framework is simple and able to show the effectiveness. However, in somewhat points, the result comes out naturally, since the ChatGPT was further fine-tuned with the augmented multi-lingual unsafe dataset. Can we conclude that the improvement depends on the extent of the low-source level? Or another insight can be concluded from the framework?
8. Lastly, the authors mention that “Additionally, the translation process enables *the transfer of knowledge and safety guidelines across multiple languages,* thereby improving the safety alignment in a multilingual context.” However, the safety guidelines can be differently applied across different cultures and societies. I am not certain that the adversarial examples used in this paper are general enough to be transferred across other languages and societies. However, at least the guidelines follow the U.S. or several companies, and I think we should carefully consider the sensitivity of safety level in the target society.

- [1] Liu et al., G-EVAL: NLG Evaluation using GPT-4 with Better Human Alignment, https://arxiv.org/pdf/2303.16634.pdf
- [2] Zheng et al., Judging LLM-as-a-Judge with MT-Bench and Chatbot Arena, https://arxiv.org/abs/2306.05685
- [3] Wang et al., Large Language Models are not Fair Evaluators, https://arxiv.org/pdf/2305.17926.pdf

**Questions:**

- In Table 1, unlike to the unintentional scenario, in the intentional scenario, it’s hard to find any tendency among HRL, MRL, and LRL.
- The generated outputs in non-English language are again translated into English to be assessed by a Human or GPT-4 evaluator, through Google Translates. In this process, why didn’t you assess the generated output itself, but translate it back to English? Do you think there could be errors?
- For simplicity, the authors set the temperature as zero for the two models. Do you think the results could vary if nucleus sampling was applied?
- “unsafe and general” phrase could cause readers to misunderstand; “general” means “safe” or “helpfulness (usefulness)” instruction pairs. Or even it can be interpreted as an unsafe prompt - a general (safe) response.

---

> ### Author Response · Authors · 2023-11-20
>
> We greatly appreciate your insightful comments and suggestions, as they have been helpful in refining and enhancing our work. We have thoroughly reviewed all of your points and have addressed your concerns as outlined below:
>
> >**W1: Standard deviations**
>
> Thank you for your kind feedback. In our experiments, we maintained a temperature of zero and utilized fixed datasets for both preliminary and detailed testing. As a result, the output remained constant. However, for supplementary experiments, we employed nucleus sampling to explore further possibilities. For more information on these supplementary experiments, please refer to Q3, where we provide detailed explanations.
>
> >**W2: Detail human evaluation process**
>
> We completely agree with this. To ensure accurate human evaluation, we have implemented the following steps:
>
> 1. Translation: As discussed in Section 2.1, we have used Google Translate to translate all the output into English.
> 2. Annotation: We engage two annotators to independently label the translated output following the definition in Appendix A.2.
> 3. Conflict resolution: In case of any conflicts or discrepancies between the annotations, we have involved an additional expert to make the final determination on the label.
>
> By following these steps, we aim to ensure the highest possible accuracy and reliability in the human evaluation process.
>
> >**W3: Address potential LLM-based evaluator bias**
>
> Thank you for bringing up this important point.
>
> However, we think that evaluator bias may mainly exist in NLG problems, particularly in subjective metrics like "fluency", "relevancy", "preference", etc. When it comes to evaluating "safety", the concept tends to be much more "objective," especially when we frame safety evaluation as a classification task.
>
> In addition, almost all examples are easy-to-identify unsafe topics, such as "Violence", "Adult content" and etc. We list the detailed statistics in Appendix A.2.
>
> Finally, we have defined the criteria for each class in the GPT-4 evaluation prompt, minimizing potential bias as much as possible. In W4, we further show that GPT-4 is a reliable evaluation measure.
>
> >**W4: Provide specific agreement values**
>
> Sure, we have measured the agreement between GPT-4 and human evaluations, utilizing Cohen's kappa coefficient. The Cohen's kappa score is 0.86, showing a high agreement between GPT-4 and human evaluations.
>
> >**W5: Publish MultiJail dataset and results**
>
> Yes, we will publish our MultiJail dataset including necessary code and results in the future.

---

> ### Author Response · Authors · 2023-11-20
>
> >**W6: Evaluate open-sourced models**
>
> Yes, we have explored evaluating open-source language models such as LLama2 and Vicuna. However, we found certain limitations in their multilingual capabilities and safety measures, which could potentially introduce noise during analysis. Specifically, LLama2 primarily responds in English, even when presented with non-English input. Though Vicuna has the ability to respond in languages other than English, it lacks specialized safety tuning, and its limited safety capabilities are mainly derived from user-shared responses of ChatGPT/GPT-4.
>
> Given these limitations, we've primarily centered our attention on ChatGPT and GPT-4. These models offer a more seamless conversational experience across various languages and have robust safety mechanisms in place. Nevertheless, we have included the experimental results for LLama2 and Vicuna under unintentional scenarios as supplementary references, presented in the table below, detailed in Appendix A.7:
>
> |              | ChatGPT | GPT-4 | Llama2-7b-Chat | Vicuna-7b-v1.5 |
> |--------------|---------|-------|----------------|----------------|
> | en - unsafe  | 0.63    | 0.95  | 0.63           | 57.14          |
> | en - safe    | 99.37   | 99.05 | 99.37          | 37.78          |
> | en - invalid | 0.00    | 0.00  | 0.00           | 5.08           |
> | HRL - unsafe | 4.34    | 3.60  | 2.22           | 40.32          |
> | HRL - safe   | 95.13   | 95.98 | 92.06          | 51.32          |
> | HRL - invalid| 0.53    | 0.42  | 5.71           | 8.36           |
> | MRL - unsafe | 11.32   | 4.13  | 4.55           | 42.75          |
> | MRL - safe   | 87.20   | 94.94 | 66.88          | 32.38          |
> | MRL - invalid| 1.48    | 0.95  | 28.57          | 24.87          |
> | LRL - unsafe | 14.92   | 10.16 | 1.69           | 28.78          |
> | LRL - safe   | 78.41   | 83.49 | 65.19          | 9.42           |
> | LRL - invalid| 6.67    | 6.35  | 33.12          | 61.80          |
> | AVG - unsafe | 14.92   | 5.96  | 2.82           | 37.28          |
> | AVG - safe   | 78.41   | 91.46 | 74.71          | 31.04          |
> | AVG - invalid| 6.67    | 2.57  | 22.47          | 31.68          |
>
> From the results, it's evident that while Llama2-Chat performs well in English, and even obtains the lowest average unsafe rate, it also generates a considerable amount of invalid output. As such, it's difficult to determine whether its low unsafe rate is a result of genuinely safe or invalid content generation.
>
> On the other hand, Vicuna demonstrated the least safety capability, with a notably high unsafe rate of 57.14% even in English. Moreover, it also had a substantial rate of generating invalid responses.
>
> In conclusion, to ensure a unbiased evaluation, we have chosen to primarily report our experiments based on ChatGPT and GPT-4. These models showcase safisfactory multilingual abilities, exhibit low invalid response rates, and have comprehensive safety tuning mechanisms in English. These factors help to isolate the language aspect during the analysis, eliminating other potential influences.
>
> >**W7: Analyze "Self-Defense" framework effectiveness**
>
> We appreciate the reviewer's insightful question. Before addressing the question, we would like to provide some context regarding the multilingual jailbreak challenge. We classify the cause of this challenge as a mismatched generalization [1], specifically the lack of a non-English corpus during safety tuning.
>
> In order to ensure the effectiveness of "Self-Defence," we believe two preconditions need to be met. Firstly, the model should possess strong safety capabilities in high-resource languages. Secondly, it should have strong multilingual abilities.
>
> If the model fulfills these two requirements, "Self-Defence" can perform explicit alignment of its safety capabilities from high-resource languages to low-resource languages. Essentially, it leverages its existing safety capability and transfers it to other language spaces.
>
> Furthermore, in our future research, we plan to conduct additional experiments to validate and enhance the effectiveness of our approach. By doing so, we aim to propose even more effective methods to tackle the multilingual jailbreak challenge.
>
> >**W8: Consider cultural differences in safety guidelines**
>
> We fully acknowledge that the concept of "safety" can vary across different cultures and societies.
>
> However, most adversarial examples consist of general unsafe content, including pornography, terrorism, fraud, and other globally recognized forms of unsafety, detailed in Appendix A.2. Consequently, only a limited number of examples are appropriate for "localization" purposes. Additionally, to focus solely on the "language" aspect, we have opted for direct translation without localization. Nonetheless, we recognize the significance of studying cultural differences, which would require fine-grained annotation and more in-depth research. We intend to explore this aspect in our future research projects.

---

> ### Author Response · Authors · 2023-11-20
>
> >**Q1: Lack of trend in intentional scenario**
>
> Yes, as discussed in Section 3.2.2, we found that LLMs display relative stability regardless of language availability in intentional scenarios. We assume that in unintentional scenarios, the multilingual jailbreak problem arises mainly due to the inherent tension between the LLMs' pre-training ability and safety capability, leading to an obvious trend. However, this situation becomes further complicated when malicious instructions are introduced, as the importance of instruction-following factors increases. There is a conflict between pre-training, instruction-following, and safety abilities. This complex situation makes the previously observed trend unobvious. We will dive deeper into our future research to study the inherent cause of the jailbreak problem.
>
> >**Q2: Translation and assessment of non-English outputs**
>
> The primary goal is to ensure the quality of evaluation. Initially, during the human evaluation process, we discovered that while the translation may not be perfect, it maintains the core idea of the sentence. This allows us to confidently predict its safety.
>
> With respect to GPT-4, the unintentional scenario results demonstrate limited safety capabilities in non-English languages. As a solution, translating the non-English output back into English using Google Translate can enhance GPT-4's safety evaluation performance. Although this may introduce some noise, the benefits it brings outweigh these minor inconveniences.
>
> In our preliminary experiments, we actually tried sampling some examples and compared GPT-4's evaluation of translated English versus direct non-English text on its website. Our investigation revealed that translating back to English is a more reliable approach.
>
> >**Q3: Zero temperature setting and nucleus sampling impact**
>
> We chose to use a zero temperature setting primarily to prioritize the reproducibility and reliability of our experiment. While other decoding strategies may yield slightly different results, a recent study [1] has indicated that the qualitative outcomes and observed trends remain consistent in general jailbreak experiments.
>
> To further investigate the impact of different decoding strategies, we conducted an experiment in an unintentional scenario using ChatGPT with nucleus sampling and a top_p value of 0.8. To ensure reliable results, we ran the experiment three times with different seeds. The following data represents the unsafe rate of this experiment, with the standard deviation values indicated within parentheses:
>
> |                        | ChatGPT (sample=False) | ChatGPT (nucleus sampling with top_p = 0.8) |
> |------------------------|------------------------|---------------------------------------------|
> | en                     | 0.63                   | 0.42 (0.18)                                 |
> | zh                     | 2.22                   | 4.02 (0.48)                                 |
> | it                     | 2.86                   | 2.75 (0.37)                                 |
> | vi                     | 7.94                   | 9.10 (0.48)                                 |
> | **HRL**                    | 4.34                   | 5.29 (0.21)                                 |
> | ar                     | 6.03                   | 6.88 (0.48)                                 |
> | ko                     | 9.84                   | 9.84 (0.84)                                 |
> | th                     | 18.10                  | 20.95 (1.45)                                |
> | **MRL**                    | 11.32                  | 12.56 (0.34)                                |
> | bn                     | 28.25                  | 31.85 (1.28)                                |
> | sw                     | 7.94                   | 8.15 (1.20)                                 |
> | jv                     | 8.57                   | 9.42 (1.43)                                 |
> | **LRL**                    | 14.92                  | 16.47 (0.60)                                |
> | **Avg.**                   | 10.19                  | 11.44 (0.31)                                |
>
> It is evident that the average unsafe rate, although slightly higher (1.25%), the trend is still clearly observable, as the rate of unsafe content increases with decreasing language availability, resulting in a consistent ranking order.
>
> >**Q4: Ambiguity of "unsafe and general" phrase**
>
> Thank you for pointing it out. "General" here primarily focuses on "usefulness". Implicitly, all content is safe. We will refine the expression to prevent misunderstandings.
>
> [1] Wei et al., Jailbroken: How Does LLM Safety Training Fail?, https://arxiv.org/abs/2307.02483
>
> [2] Ganguli et al., Red teaming language models to reduce harms: Methods, scaling behaviors, and lessons learned, https://arxiv.org/abs/2209.07858

---

> ### Comment · Reviewer_emGp · 2023-11-22
> **Thanks for responding my reviews.**
>
> Dear authors,
>
> I thoroughly read your responses, and thank you for updating your manuscript and providing additional experiments.
> I was able to resolve my questions. Followings are my thoughts and comments:
>
> 1. W6: Evaluate open-sourced models
> * I understand the reason why you chose GPT-4 and ChatGPT. From the additional experiments, however, the trend where low resource language is vulnerable to jailbreaks does not work. But, when it comes to "safe" instead of "unsafe", that is "unsafe + invalid", we can conclude that the extent of safe response is decreasing as the volume of resource is decreasing.
>
> 2. Q3: Zero temperature setting and nucleus sampling impact
> * I appreciated your supplementary experiments. I think even though zero-temperature, the probability distribution probably and slightly different because of randomness of layers in the LLM. But, again, I understand!
>
> 3. W7: Analyze "Self-Defense" framework effectiveness
> >
> >In order to ensure the effectiveness of "Self-Defence," we believe two preconditions need to be met. Firstly, the model should possess strong safety capabilities in high-resource languages. Secondly, it should have strong multilingual abilities.
> >
> >If the model fulfills these two requirements, "Self-Defence" can perform explicit alignment of its safety capabilities from high-resource languages to low-resource languages. Essentially, it leverages its existing safety capability and transfers it to other language spaces.
> >
>
> I was able to understand the authors' motivation for "self-defense". I'd recommend to mention and explain the motivation and assumptions in the manuscript. However, to verify and support them, you might need the future work you mentioned.
>
> 4. (minor) The algorithm 1 looks confusing.
> - please clarify definition of D_l,
> - "M: D_a <- M(D_s)" could be understated as the definition of a function M
> - in line 1, how to augment? what does it mean that D_a <- M(D_s)? That is, M(D_s) = set of responses to queries in D_s?
>
> Good luck :)

---

> > ### Author Response · Authors · 2023-11-22
> >
> > Dear Reviewer emGP,
> >
> > Thank you for your prompt reply and additional valuable feedback!
> >
> > Here are our responses to your comments:
> >
> > >**1. W6: Evaluate open-sourced models**
> >
> > Yes, the inherent limitations of the two open-source LLMs may compromise the validity of the evaluation. To ensure a balanced and unbiased assessment, we have opted to use GPT-4 and ChatGPT.
> >
> > >**2. Q3: Zero temperature setting and nucleus sampling impact**
> >
> > Yes, achieving a completely deterministic result can be challenging. However, we strive to enhance the reproducibility of our experiments by setting the temperature to 0.
> >
> > >**3. W7. Analyze "Self-Defence" framework effectiveness**
> >
> > Thank you for your constructive feedback! We appreciate your valuable input and will carefully consider it to improve our manuscript.
> >
> > >**4. W7. Analyze "Self-Defence" framework effectiveness**
> >
> > 1. "D_l" in this context refers to the translated dataset in the target languages.
> >
> >  2&3. We aim to showcase the augmentations performed by the LLM using these seed examples as partial input.
> >
> > We will refine our manuscript and provide clear definitions to avoid any misunderstandings.
> >
> > As we have taken care of your concerns, would you be open to discussing a potential rate change? We would greatly appreciate your consideration. Again, thank you for your helpful comments! Your valuable input has greatly improved our manuscript.
> >
> > Best Regards,
> >
> > Authors

---

> > > ### Comment · Reviewer_emGp · 2023-11-22
> > > **Change my score**
> > >
> > > I changed my overall scores from 6 to 8 by acknowledging the important topic of this paper and its value as benchmark baseline of multi-lingual jailbreak.
> > >
> > > Best regard,

---

### Author Response · Authors · 2023-11-20
**General Response to All Reviewers**

We are deeply grateful for the valuable time and dedication put forth by each reviewer in providing insightful and constructive feedback. We have now uploaded a revised version of the paper that includes additional details, and supplementary experiments to enhance clarity.

To sum up, we have made the following revisions:
- Additional descriptions for Figure 2 to quantify the agreement between human and GPT-4 evaluation;
- Additional tag statistics to illustrate safety issues covered in the dataset in Appendix A.2;
- Supplementary experiment in Appendix A.6 to study the effect of decoding methods;
- Supplementary experiment in Appendix A.7 to show the results of open-source LLMs;
- Additional explanation of selecting AIM in Section 3.1;
- Additional description on the flexibility of Self-Defence in Section 4.1;
- Additional description on the effect of noisy translation in Section 4.3.

We addressed individual reviewer concerns below. Thank you all once again!

---

### Meta-Review · Area_Chair_rr3K · 2023-12-06

**Metareview:**

This paper investigates the vulnerability of large language models (LLMs) to multilingual jailbreaks in both unintentional and intentional scenarios. It proposes a self-defense framework to improve LLM safety across languages.

Strengths:

- Addresses a timely and crucial issue.
- Identifies multilingual vulnerabilities and evaluates the impact of language resource levels.
- Introduces the MultiJail dataset for multilingual jailbreak research.
- Proposes the SELF-DEFENSE framework for improving multilingual safety.

Weaknesses:

- Presentation could be improved with details.
- Self-defense framework needs comparison to other safety techniques and might be overly complicated.
- Potential bias in using GPT-4 as an evaluation measure.

**Justification For Why Not Higher Score:**

Evaluation can be further improved (soundness, presentation details). Few reviewers also pointed out the the mitigation section is relatively weak.

**Justification For Why Not Lower Score:**

Addresses the crucial issue of LLM safety in non-English languages.

---

### Decision · Program_Chairs · 2024-01-16

Accept (poster)